# PLUM: Improving Code LMs with Execution-Guided On-Policy Preference Learning Driven By Synthetic Test Cases

## Abstract

Preference learning provides a promising solution to address the limitations of supervised fine-tuning (SFT) for code language models, where the model is not explicitly trained to differentiate between correct and incorrect code. Recent findings demonstrate that on-policy data is the key to successful preference learning, where the preference data is collected using the same policy LM being trained. Inspired by this, we propose PLUM, an on-policy **P**reference **L**earning framework A**u**gmented with test cases for code L**M**s. The framework operates in three key stages: (1) automatic generation of test cases from natural language instructions, (2) creation of a preference data by evaluating candidate code solutions sampled from the policy, which can then be used to (3) train the policy LM. PLUM levitates the need to train reward models, allowing for large scale on-policy and online preference data collation. PLUM is evaluated on both standard benchmarks (HumanEval, MBPP) and more challenging ones (LiveCodeBench), delivering substantial improvements over original SFT'ed models and other execution-feedback-driven approaches. We show PLUM's benefits are consistent across various widely-used code LMs even they have been well-trained with SFT. For example, PLUM increases pass rates by up to 4.8% on average on standard benchmarks and 11.8% on LiveCodeBench, demonstrating its effectiveness and generalizability. We also demonstrate the benefits of on-policy and online preference learning by comprehensive experimentation.

## 1 Introduction

Language models pre-trained on code corpora have excelled at code generation (Rozière et al., 2024; Li et al., 2023). Supervised Fine-Tuning (SFT) enhances their ability to follow natural language prompts but focuses on reproducing patterns from training data rather than ensuring code correctness (Wei et al., 2023; Zheng et al., 2024). This leads to models that generate syntactically correct but functionally flawed code, unable to meet real-world requirements like edge cases or algorithmic accuracy (Chen et al., 2024). Works like AlphaCode (Li et al., 2022) and LeTI (Wang et al., 2024b) have introduced test outcomes as a means to define functional correctness in code generation. Building on the insights from these efforts, we propose leveraging preference learning for refining model behavior. Preference learning trains models to prefer certain solutions (e.g., factual, helpful, or harmless) over undesirable ones (e.g., inaccurate, unhelpful, or harmful). Despite its success in aligning models with human values and improving reasoning in other domains (Dong et al., 2023; Guo et al., 2024a; Yuan et al., 2024; Wang et al., 2024a; Pang et al., 2024), the application of preference learning as a principled and efficient approach in code generation remains under-explored, largely due to the lack of high-quality training data.

Recent research shows that reducing the likelihood of incorrect outputs is more effective for improving model performance than simply maximizing correct responses (Setlur et al., 2024; Tajwar et al., 2024). Mode-seeking objectives, which prioritize minimizing errors, have been found to outperform maximum likelihood methods by more efficiently redistributing probability mass across potential outputs. This underscores the importance of applying on-policy and online approaches to enhance preference learning algorithms (Tajwar et al., 2024; Guo et al., 2024b; Setlur et al., 2024; Liu et al., 2024). Unlike offline preference data, on-policy data remains in-distribution with the model, reducing the risk of misalignment (Guo et al., 2024b; Zhang et al., 2024; Tang et al., 2024; Fisch et al., 2024). The main challenge now is how to efficiently obtain preference labels for on-policy data at scale (Yang et al., 2024b). In programming tasks, test cases present as a native and powerful candidate solution to

address this issue. Being able to automatically produce high-quality test cases unlocks the possibility of collecting preference data over programming questions at any scale.

---

**Prompt for Test Case Generation**

You are a teaching assistant helping to write reference solutions and tests for programming questions. Given a programming question, you need to first analyze the problem, then write a reference solution (code), followed by assertions that test student solutions. The test code must be runnable when concatenated at the end of student solutions to check the correctness.

**Programming Question:**
`{Question}`

Follow the format below:
**[Analysis]**
`{{Natural language analysis of the problem.}}`
**[Solution]**
`{{Your solution to the problem}}`
**[Start Code]**
`{{Start code for students so that they can follow the I/O protocol.`
`E.g. Function signatures, class names etc.}}`
**[Test Code]**
`{{Test code that is immediately runnable if concatenated with`
`student code to check the correctness.}}`

---

To this end, we propose **p**reference **l**earning framework a**u**gmented with test cases for training code language **m**odels (**PLUM**), which integrates the process of deriving test cases from natural language specifications into the training process to obtain preference labels for model's on-policy candidate solutions. PLUM utilizes natural language instructions from well-established datasets such as OSS-Instruct (Wei et al., 2023), Evol-Instruct-Code (Luo et al., 2023), and ShareGPT (Cleston, 2023). For each instruction, high-quality test cases are constructed, and multiple solutions are sampled from the model. These solutions are evaluated using the generated test cases, with preference labels assigned based on the results: solutions passing the tests are preferred, while failures are dis-preferred. This dataset then trains the policy using established preference learning algorithms (Rafailov et al., 2023; Ethayarajh et al., 2024; Azar et al., 2023). By relying solely on policy model's self-generated solutions, PLUM eliminates the need for external, synthetic off-policy data, reducing the risk of distributional shifts and poor generalization commonly observed for synthetic off-policy data, enhancing the model's robustness and improving its ability to differentiate between correct and incorrect solutions. In addition, by showcasing the effectiveness of our framework, we demonstrated the feasibility of bypassing tedious (and potentially unstable) reward model training (Liu et al., 2023a; Le et al., 2022) and manual labeling, by automating the process of test case synthesis. These simplicity advantages of PLUM makes online preference training of language models possible.

We evaluate PLUM on a diverse set of state-of-the-art code language models under different set-ups, on commonly used benchmarks: HumanEval(+) and MBPP(+) (Chen et al., 2021; Austin et al., 2021; Liu et al., 2023b) as well as more challenging code generation datasets like LiveCodeBench and LeetCode (Jain et al., 2024; Guo et al., 2024a). We demonstrate that our approach seamlessly integrates with various models in a plug-and-play manner, relying solely on coding instructions to enhance models' code generation capabilities. Furthermore, we show that online training, facilitated by automated test case generation, further boosts model performance particularly on difficult coding benchmarks, echoing findings from other domains (Xiong et al., 2024b).

## 2 PREFERENCE LEARNING AUGMENTED WITH TEST CASES FOR CODE LMs (PLUM)

The core of PLUM lies in leveraging recent advancements in on-policy and online preference learning, which have proven effective across various domains (Xiong et al., 2024b;a; Mitra et al., 2024). In the context of code generation, PLUM simplifies and scales the preference data collection process

by using test cases. These test cases act as a lightweight yet robust mechanism to evaluate model outputs.

Algorithm 1 outlines the core mechanism of PLUM, demonstrating how test case generation is embedded into the preference learning loop. This process allows model-generated outputs to be evaluated in real-time using automatically generated test cases, which serve as direct feedback mechanisms for the learning process. By using test cases rather than complex reward models, PLUM simplifies the collection of preference data, maintaining high feedback quality while reducing the complexity associated with reward model training.

---

**Algorithm 1** PLUM.

---

**Input:** Natural language instructions $\mathcal{I} = \{q_i\}$, policy model to be trained $\pi_\theta$, generator model $G$, update frequency $T$, chunk size $M$      ▷ Unified for both offline and online alignment
**Output:** Trained policy model $\pi'_\theta$
1:   Initialize preference datasets $\mathcal{D}^+$ and $\mathcal{D}^-$
2:   **for** each chunk $\mathcal{I}_M \subset \mathcal{I}$, where $\mathcal{I}_M$ contains $M$ instructions **do**
3:      **for** each $q_i \in \mathcal{I}_M$ **do**
4:         Generate $n$ pairs of reference code and test case $\{(r_{ij}, t_{ij})\}_{j=1}^n$ using $G$    ▷ Test collection
5:         **for** each pair $(r_{ij}, t_{ij})$ **do**
6:            **if** $r_{ij}$ passes $t_{ij}$ **then**
7:              Add $(q_i, t_{ij})$ to $\mathcal{D}$    ▷ Self-consistency filtering
8:            **end if**
9:         **end for**
10:        $\mathcal{S}_{ik} \sim \pi_\theta$ for $k = 1$ to $K$ (sample $K$ solutions for $q_i$)    ▷ On-policy sampling
11:        **for** each solution $s_{ik} \in \mathcal{S}_{ik}$ **do**
12:           **for** each test case $t_{ij}$ in $\mathcal{D}$ **do**
13:              **if** $s_{ik}$ fails $t_{ij}$ **then**
14:                 Add $(q_i, s_{ik})$ to $\mathcal{D}^-$    ▷ Negative case
15:              **end if**
16:           **end for**
17:           Add $(q_i, s_{ik})$ to $\mathcal{D}^+$    ▷ Positive case
18:        **end for**
19:      **end for**
20:      Filter out instances with no correct solutions from $\mathcal{D}^+$ and $\mathcal{D}^-$
21:      **if** iteration count $\% T = 0$ **then**    ▷ Policy Update
22:         Train the policy model $\pi_\theta$ using $\mathcal{D}^+$ and $\mathcal{D}^-$ with preference learning to get $\pi'_\theta$
23:         Update policy model $\pi_\theta = \pi'_\theta$
24:      **end if**
25: **end for**
26: **return** $\pi'_\theta$

---

## 2.1 THE PLUM

As illustrated in Algorithm 1 and Figure 1, our approach takes in a base policy model, a set of natural language programming instructions, and a test case generator. For each iteration, it produces multiple test cases for the batch of instructions. Then we sample solutions from policy $\pi_\theta$, and execute them against the generated test suite to obtain preference labels. We then update the policy $\pi_\theta := \pi'_\theta$.

## 2.2 GENERATING TEST CASES

A crucial factor in making PLUM successful is the ability to synthesize high-quality test cases for programming questions. In the following subsections, we provide a detailed explanation of the test case generation process, outlining how it contributes to the overall effectiveness of PLUM.

The test cases in PLUM are generated with a test-case generator model over natural instructions from established code generation datasets.[1] In automated testing, ensuring the correctness and completeness of test cases is a persistent challenge due to the lack of reliable oracles to validate test outputs. We adopt two strategic principles: 1) employing self-consistency as an approximate

---

[1] We use GPT-4-1106 as the generator model.

oracle, and 2) generating diverse test suites to minimize overfitting to any particular test instance and mitigate under-specification.

**Collecting instructions from established datasets**   We collect natural language instructions from established datasets including OSS-Instruct (Wei et al., 2023), Evol-Instruct-Code (Luo et al., 2023), and ShareGPT (Cleston, 2023). [2]These datasets provide a diverse range of programming tasks and instructions. Although they come with gold/silver solutions in the training splits, these solutions are *never* used in PLUM. Instead, they allow us to directly compare PLUM's performance against SFT, which relies on gold solutions, as we investigate in our experiments. Not requiring gold solutions for training broadens the applicability of PLUM to a wide variety of real-world coding tasks and user requirements.

**Generating high-quality test cases**   Given a training instruction in natural language, we prompt a generator model to produce a reference solution, a starter code snippet specifying the function signature, and a suite of test cases using the prompt in Figure 1. The generated test cases are critical for ensuring that the solutions meet the functional requirements specified in the instructions. The correctness of the test cases is central to the success of preference learning. We adopt a consistency-based approach inspired by Chen et al. (2023) and Rozière et al. (2024) for quality control.

We check for consistency between the generated reference solution and the test cases. Pairs where the test cases do not accurately reflect the solution, or the solution does not pass the test cases, are filtered out. This process helps minimize potential noise and enhances the quality of the test cases used in the following stages. The generated reference solutions serve only to control the quality of the test cases and are *never* used in training. Similarly, the solutions provided with the instruction data are *never* used in PLUM. On average, each instruction is paired with 3–5 test cases depending on the dataset.

## 2.3 SAMPLING SOLUTIONS FROM THE POLICY TO CREATE THE PREFERENCE DATA

Many preference learning algorithms assume that the preference data is in-distribution for the policy, i.e., the solutions are sampled from the policy model to be trained Rafailov et al. (2023); Ethayarajh et al. (2024); Azar et al. (2023). In practice, however, preference data often contains solutions sampled from different models than the policy, leaving the data out of distribution Bai et al. (2022); Yuan et al. (2024). A common workaround is to first perform supervised fine-tuning (SFT) on the same instructions before applying preference learning (Rafailov et al., 2023; Yuan et al., 2024). This ensures that the policy has a similar distribution to that from which the preference data are sampled.

One of the research questions we aim to answer through PLUM is the standalone effect of preference learning on LMs' coding capability, with or without first performing SFT. To this end, we sample solutions from the policy to be trained and run them against the test cases to create the preference data. For each instruction, we sample $K$ solutions from the policy and evaluate them against the generated test cases. $K$ is set to 20 based on the findings

| Dataset | Self-Consistency Pass Rate(%) |
|---|---|
| OSS-INSTRUCT | 63.76 |
| EVOL-INSTRUCT | 42.38 |
| SHAREGPT | 45.69 |

Table 1: Self-Consistency Pass Rate Using GPT-4-1106.

from our preliminary experiments. With static and execution checks,[3] we identify and filter out solutions that contain syntactic errors and fail to execute, as our focus is on functional correctness.

Moreover, as a recent work points out, training with code snippets containing syntax errors may hurt the model's performance (Wang et al., 2024b). Solutions passing all test cases are used as the chosen solutions, and those failing at least one the rejected solutions.

An instruction is filtered out if it has no chosen solution after this process.

This aims to ensure that the learned policy does not drift too far from the original one as drastic changes might cause the model to forget previously learned information or to perform poorly on tasks it was previously adept at Rafailov et al. (2023).

---

[2]We focus on Python due to its wide use and the availability of well-established training and evaluation resources.

[3]We use mypy for the static check: `https://mypy-lang.org/`.

| Model | Item | MBPP | MBPP+ | HE | HE+ | Avg. |
|---|---|---|---|---|---|---|
| | Baseline | 77.7 | 67.2 | 83.5 | 78.7 | 76.8 |
| | Cond.Token | 71.9 | 57.1 | 68.3 | 57.9 | 63.8 |
| | RFT | 81.2 | 67.9 | 84.1 | 81.1 | 78.6 |
| CODEQWEN-1.5-CHAT (Bai et al., 2023) | Cond.Err.Msg | 79.4 | 68.7 | 84.1 | 79.3 | 77.9 |
| | PLUM-DPO | 81.2 | 70.2 | 86.0 | 81.1 | **79.6** |
| | **PLUM-KTO** | 81.0 | 69.0 | 86.0 | 81.1 | 79.3 |
| | *Rel. +* | *4.3* | *2.7* | *3.0* | *3.1* | *3.3* |
| | Baseline | 74.9 | 65.6 | 75.4 | 71.3 | 71.8 |
| | Cond.Token | 73.4 | 62.9 | 76.8 | 70.7 | 71.0 |
| | RFT | 74.7 | 64.9 | 74.4 | 66.5 | 70.1 |
| DS-CODER-INSTRUCT (Guo et al., 2024a) | Cond.Err.Msg | 74.7 | 64.2 | 80.5 | 75.0 | 73.6 |
| | PLUM-DPO | 76.4 | 65.9 | 80.5 | 76.8 | 77.4 |
| | **PLUM-KTO** | **78.2** | **67.9** | **81.7** | **76.8** | **76.2** |
| | *Rel. +* | *4.4* | *3.5* | *8.4* | *7.7* | *6.0* |
| | Baseline | 75.4 | 61.9 | 66.5 | 60.4 | 66.1 |
| | Cond.Token | 75.9 | 62.4 | 68.3 | 62.2 | 67.2 |
| | RFT | 76.2 | 62.2 | 67.7 | 62.2 | 67.1 |
| MAGICODER-DS (Wei et al., 2023) | Cond.Err.Msg | 74.2 | 62.2 | 66.5 | 59.8 | 65.7 |
| | PLUM-DPO | 75.9 | 63.7 | 67.7 | 61.6 | 67.2 |
| | **PLUM-KTO** | **79.6** | **66.7** | **71.3** | **65.9** | **70.9** |
| | *Rel. +* | *5.6* | *7.8* | *7.2* | *9.1* | *7.4* |
| | Baseline | 75.7 | 64.4 | 76.8 | 70.7 | 71.9 |
| | Cond.Token | 73.9 | 63.7 | 75.0 | 71.3 | 71.0 |
| | RFT | 75.4 | 64.4 | 73.2 | 69.5 | 70.6 |
| MAGICODER-S-DS (Wei et al., 2023) | Cond.Err.Msg | 75.2 | 65.4 | 75.6 | 70.7 | 71.7 |
| | PLUM-DPO | 76.2 | 64.7 | 78.7 | 73.8 | 73.4 |
| | **PLUM-KTO** | **80.4** | **69.3** | **80.5** | **73.8** | **76.0** |
| | *Rel. +* | *4.4* | *7.4* | *4.4* | *4.5* | *5.2* |
| | Baseline | 73.9 | 63.7 | 77.4 | 72.0 | 71.8 |
| | Cond.Token | 73.9 | 62.9 | 75.6 | 71.3 | 70.9 |
| | RFT | 74.2 | 63.7 | 76.8 | 72.0 | 71.7 |
| OCI-DS (Zheng et al., 2024) | Cond.Err.Msg | 75.0 | 70.7 | 74.4 | 64.7 | 71.2 |
| | PLUM-DPO | 76.4 | 66.4 | 80.5 | 76.2 | 74.9 |
| | **PLUM-KTO** | **78.2** | **66.4** | **80.5** | **76.2** | **75.3** |
| | *Rel. +* | *5.8* | *4.2* | *4.0* | *5.8* | *5.0* |
| | Baseline | 66.4 | 55.4 | 72.6 | 65.2 | 64.9 |
| | Cond.Token | 59.6 | 48.4 | 23.2 | 21.3 | 38.1 |
| | RFT | 63.2 | 52.6 | 60.4 | 56.7 | 58.2 |
| OCI-CL (Zheng et al., 2024) | Cond.Err.Msg | 67.9 | 55.9 | 68.9 | 65.2 | 64.5 |
| | PLUM-DPO | 66.4 | 55.9 | 71.3 | 65.2 | 64.7 |
| | **PLUM-KTO** | **66.7** | **55.4** | **73.8** | **67.7** | **65.9** |
| | *Rel. +* | *0.5* | *0.0* | *1.7* | *3.8* | *1.5* |

Table 2: %Pass@1 on HumanEval (HE) and MBPP, and their enhanced versions (HE+ and MBPP+) when PLUM is applied to OSS-Instruct. The *Rel. +* is computed as the relative percentage increase of PLUM-KTO over baseline. PLUM brings consistent improvements over SFT-ed baseline and outperforms other methods that leverage execution feedback when applied to the same SFT-ed models.

## 2.4 PREFERENCE LEARNING

We then proceed to train the model on the on-policy sampled candidate solutions using preference learning algorithm. In this process, we do not need golden solutions paired in the original dataset or GPT-4 during test-generation process. We mainly consider two popular preference learning algorithms - Direct Preference Optimization Rafailov et al. (2023) and Kahneman-Tversky Optimization (Ethayarajh et al., 2024) that have been shown to bring improvements for reasoning tasks (Mitra et al., 2024; Yuan et al., 2024; Dubey et al., 2024). For DPO, we subsample redundant classes and randomly pair positive and negative responses for each programming question. In contrast, we use all available responses when training with KTO.

Figure 1: Overview of PLUM. It involves three steps: (1) Generating the test cases; (2) Sampling solutions from the policy and evaluating them against the test cases to collect the preference data for (3) preference learning.

## 3 EXPERIMENTS

To demonstrate the effectiveness of PLUM, we evaluate it on established benchmarks: HumanEval (Chen et al., 2021), MBPP (Austin et al., 2021), and EvalPlus (a widely-adopted augmented version (Liu et al., 2023b) of them). We also use the more challenging LiveCodeBench (Jain et al., 2024).

**Datasets** To demonstrate the generality of our approach across different datasets, we evaluated it on three distinct collections of open datasets: OSS-Instruct (GPT-3.5 generated), EvolInstruct, and a Python code generation subset of ShareGPT. We showed that PLUM can significantly enhance model performance in various settings with high data efficiency, even when using only a small, randomly selected subset of SFT datasets.

**Preference Data Collection** We use powerful language models to generate test cases for each programming question. The results reported in the main text use test cases generated by GPT-4 OpenAI et al. (2024). An ablation of test case generators is in Appendix A.5. For the OSS-Instruct dataset and ShareGPT dataset, we query GPT-4 for 3 responses for a randomly chosen subset of 1500 questions, and due to the comparatively more complex nature of the natural language instruction, we generate 6 for each of the EvolInstruct instances for a subset of 1000.

We then sample 20 outputs from the policy using temperature $T = 1$ for the former two and 50 outputs for the latter. This yields around $\sim 60,000$ examples for ShareGPT and OSS-Instruct, and around $\sim 120,000$ for EvolInstruct before any filtering. We present the statistics on the pass ratio of sampled solutions over OSS-Instruct in Figure 3. We included the self-consistency pass rate of the test-generation process with GPT-4-1106 in Table 1.

| Model | Item | MBPP/(+) | HE/(+) |
|---|---|---|---|
| DS-CODER -INSTRUCT | Base | 75/66 | 75/71 |
| | Reflexion | 34/- | 43/- |
| | **PLUM** | **78/68** | **82/77** |
| CODEQWEN -7B-CHAT | Base | 78/67 | 84/79 |
| | Reflexion | 74/- | 83/- |
| | **PLUM** | **81/69** | **86/81** |

Table 3: Comparison with Reflexion (Shinn et al., 2023)

**Models** We consider a diverse set of strong open language models: MagiCoder (Wei et al., 2023), OpenCodeIntepreter (Zheng et al., 2024), CodeQwen (Bai et al., 2023), DeepSeek Coder (Guo et al., 2024a) and StarCoder2 (Li et al., 2023). MagiCoder and OpenCodeIntepreter contain instruction-tuned checkpoints from DeepSeek Coder and CodeLlama (Rozière et al., 2024) base models. In the main text, we focus on instruction-tuned language models, while differing results from directly training base models to Appendix A.4

**Baselines** We evaluate our approach against a variety of baselines, including both prompting-based and fine-tuning techniques. To compare methods that incorporate program correctness through execution feedback, we benchmark our approach against *Reflexion* (Shinn et al., 2023), using

| Model Families | Data | Item | MBPP | MBPP+ | HE | HE+ | Avg. |
|---|---|---|---|---|---|---|---|
| WIZARDCODER (Luo et al., 2023) | EVOLINSTRUCT | Baseline | 48.2 | 40.9 | 56.6 | 47.1 | 48.2 |
| | | **PLUM-KTO** | **54.3** | **48.8** | **65.9** | **52.9** | **55.5** |
| | | *Rel. +* | *16.4* | *12.3* | *12.7* | *19.3* | *15.2* |
| DS-CODER-INSTRUCT | | Baseline | 74.9 | 65.6 | 75.4 | 71.3 | 71.8 |
| | | **PLUM-KTO** | **77.7** | **67.7** | **81.7** | **76.8** | **76.0** |
| | | *Rel. +* | *3.7* | *3.2* | *8.4* | *7.7* | *5.8* |
| DS-CODER-INSTRUCT | SHAREGPT-PYTHON | Baseline | 74.9 | 65.6 | 75.4 | 71.3 | 71.8 |
| | | **PLUM-KTO** | **79.2** | **67.9** | **77.4** | **73.8** | **74.6** |
| | | *Rel. +* | *5.7* | *3.5* | *2.8* | *3.5* | *3.9* |
| OCI-DS | | Baseline | 73.9 | 63.7 | 77.4 | 72.0 | 71.8 |
| | | **PLUM-KTO** | **77.7** | **64.4** | **79.9** | **75.6** | **74.4** |
| | | *Rel. +* | *5.1* | *1.1* | *3.2* | *5.0* | *3.6* |
| CODEQWEN-1.5-CHAT | | Baseline | 77.7 | 67.2 | 83.5 | 78.7 | 76.8 |
| | | **PLUM-KTO** | **81.2** | **69.7** | **85.4** | **79.3** | **78.9** |
| | | *Rel. +* | *4.5* | *3.7* | *2.3* | *0.8* | *2.8* |

Table 4: PLUM on other datasets.

instruction-tuned (SFT) models and value-conditioning techniques (Li et al., 2022; Wang et al., 2024b). Additionally, to contrast preference-learning techniques with the SFT approach, we perform experiments using rejection-sampling-based SFT, utilizing the same set of positive examples as used in KTO. In these experiments, the solutions are also generated on-policy.

## 3.1 TRAINING

In order to demonstrate the generality of the approach when applied to various models and code instruction tuning data distributions, we experimented with different data-model pairs. We followed the procedure described earlier in the paper, and used all positive and negative responses when training with KTO objective.

## 3.2 RESULTS

**HumanEval(+) and MBPP(+)**   Table 2 presents the results of PLUM when applied to a subset of 1K instances of the OSS-Instruct-75K dataset.

MagiCoder models (-DS, -S-CL, and -S-DS) and OpenCodeIntepreter models (-CL and -DS) have already seen these instructions during supervised fine-tuning, while DeepSeekCoder-Instruct has not, as it was released earlier than the dataset. CodeQwen chat model uses proprietary data. PLUM-ShareGPT data for preference learning is generated with the same setting.  Similarly, Table 4 corresponds to the results when we apply PLUM to EvolInstruct (Luo et al., 2023) dataset. Since the instructions are comparatively less clear than the OSS-Instruct dataset, we control the number of initial samples to be the same by generating 50 samples for each problem and use about 400 instances in total.

PLUM consistently improves the performance of a wide range of code language models across all three settings, regardless of the base models' performance. Remarkably, PLUM can even improve the state-of-the-art 7B model, CodeQwen-7B-Chat, relatively by 3% on average, using either OSS-Instruct or ShareGPT data. These results demonstrate that PLUM is broadly applicable in different datasets and settings.

We noticed that PLUM-KTO consistently out-performs the baseline, and that PLUM-DPO sometimes under-perform PLUM-KTO. Prior works (Mitra et al., 2024; Yuan et al., 2024) noticed the phenomenon where DPO can exhibit instability due to reducing reward for the positive class.

**LiveCodeBench**   We further evaluate PLUM using strong instruction-tuned models on the more challenging LiveCodeBench dataset. As shown in Table 5, the models demonstrate overall performance improvements over their respective baselines across the board.  Despite the increased difficulty and reasoning required, we show that PLUM can enhance the base models' overall coding performance on interview-level coding problems from LiveCodeBench.

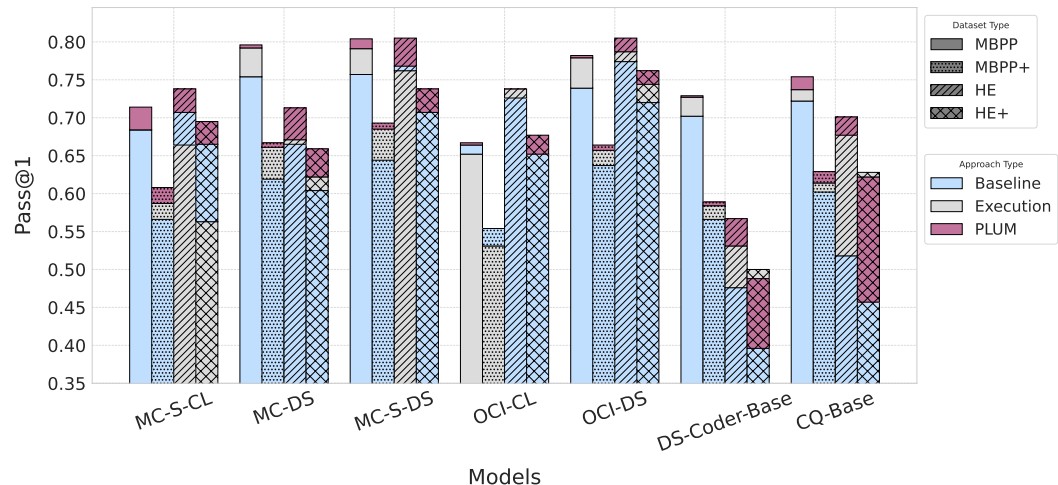

Figure 2: Ablation studies on preference training signals show that merely using un-runnable code as negative instances does not consistently enhance performance. In contrast, PLUM effectively improves the model by introducing functional correctness signals. Baseline results refer to the SFT model without PLUM.

| Model | Item | Easy | Medium | Hard | Overall |
|---|---|---|---|---|---|
| MAGICODER-S-CL | Baseline | 29.9 | 1.0 | 0 | 11.4 |
| | **PLUM-KTO** | **38.1** | **1.8** | **0** | **14.3** |
| | *Rel. +* | *27.4* | *80* | *0* | *25.4* |
| MAGICODER-DS | Baseline | 35.2 | 3.6 | 0.0 | 14.0 |
| | **PLUM-KTO** | **55.6** | **13.1** | **2.2** | **25.8** |
| | *Rel. +* | *58.0* | *266.7* | *-* | *83.9* |
| MAGICODER-S-DS | Baseline | 48.6 | 12.1 | 0.1 | 22.6 |
| | **PLUM-KTO** | **52.1** | **15.5** | **0.1** | **25.0** |
| | *Rel. +* | *7.2* | *28.1* | *0.1* | *10.6* |
| OCI-DS | Baseline | 49.6 | 9.9 | 0.4 | 21.9 |
| | **PLUM-KTO** | **45.8** | **13.7** | **1.2** | **22.3** |
| | *Rel. +* | *-7.7* | *38.4* | *200* | *1.8* |
| CODEQWEN-1.5-CHAT | Baseline | 42.7 | 18.8 | 0.9 | 23.2 |
| | **PLUM-KTO** | **43** | **23.2** | **3** | **25.8** |
| | *Rel. +* | *0.7* | *20* | *230* | *11.2* |

Table 5: %Pass@1 on LiveCodeBench.

PLUM proves particularly beneficial for medium-level interview questions, which are often quite challenging for models, especially those with around 7B parameters. This demonstrates that PLUM does more than simply fitting to commonly tested benchmarks; it enhances the models' general coding capabilities in more complex and diverse coding scenarios.

**Comparison Against Baselines**   As shown in Table 3, verbal reinforcement learning like Reflexion (Shinn et al., 2023), does not perform well on code language models fine-tuned for code at the scale relevant to our work. This is partly due to the limitations of smaller LLMs in handling various types of instructions effectively.

Approaches like LeTI (Wang et al., 2024b) implicitly optimizes the model for generating correct programs solely through input prompt, without directly enforcing such distinction with its training objective. Unlike preference learning algorithms ,approaches like LeTI Wang et al. (2024b) optimize models to generate correct programs based solely on the input prompt, without explicitly enforcing correctness through the training objective. As a result, we observe inconsistent outcomes when applying these methods to our tested SFT models, as shown in Table 2. Additionally, our results show that using value-token-conditioned approaches often lead to reduced performance, likely due to the token addition to tokenizer and the absence of clear labels distinguishing good from bad outputs.

| | On-Policy | Off-Policy | | | | |
|---|---|---|---|---|---|---|
| | **PLUM** | **DS-Coder -1.3B -Instruct** | **Qwen2.5- Coder-1.5B -Instruct** | **Codestral -22B** | **DeepSeek -Coder-33B -Instruct** | **Syn. -Neg.** |
| CODEQWEN1.5-CHAT | **79.3** | 78.4 | 78.1 | 77.3 | 79.0 | 73.9 |
| DS-CODER-INSTRUCT | **76.2** | 76.0 | 75.2 | 74.6 | 74.0 | 69.4 |
| MAGICODER-DS | **70.9** | 68.4 | 68.6 | 68.3 | 67.0 | 61.2 |
| MAGICODER-S-DS | **76.0** | 75.8 | 75.3 | 74.1 | 73.8 | 71.6 |

Table 6: Comparison between on-/off-policy preference learning using KTO on CODEQWEN-1.5-CHAT.

| | **Easy** | **Medium** | **Hard** | **Average** | **Rel. Gain** |
|---|---|---|---|---|---|
| CODEQWEN1.5-CHAT | 66.7 | 27.5 | 13.6 | 33.9 | - |
| +PLUM-DPO | 66.7 | 31.9 | 15.9 | 36.7 | 8.3 |
| **+PLUM-DPO-Iter** | **66.7** | **31.9** | **18.2** | **37.2** | **9.7** |
| +PLUM-KTO | 68.9 | 30.8 | 11.4 | 35.6 | 5.0 |
| **+PLUM-KTO-Iter** | **66.7** | **31.9** | **18.2** | **37.2** | **9.7** |
| MAGICODER-DS | 33.3 | 17.6 | 9.1 | 19.4 | - |
| +PLUM-DPO | 44.4 | 15.4 | 13.6 | 22.2 | 14.4 |
| **+PLUM-DPO-Iter** | **46.7** | **17.6** | **11.4** | **23.3** | **20.1** |
| +PLUM-KTO | 48.9 | 16.5 | 9.1 | 22.8 | 17.5 |
| **+PLUM-KTO-Iter** | **44.4** | **17.6** | **11.4** | **22.8** | **17.5** |

Table 7: Results on iterative PLUM following Algorithm 1.

**On-Policy vs Off-Policy** We now address whether "on-policy" training makes a significant difference. To test this, we apply the same method but use preference data sampled from other models (i.e., off-policy). We selected models of varying sizes, including DeepSeek-Coder-1.3B-Instruct, Qwen-2.5-Coder-1.5B-Instruct, CodeStral-22B (MistralAI, 2024), and DeepSeek-Coder-33B-Instruct. We followed the exact same data collection and training processes. Our results show that while off-policy training still improves model performance, it generally underperforms compared to on-policy training. These findings are consistent with prior research (Xiong et al., 2024a; Dong et al., 2024; Tajwar et al., 2024; Xu et al., 2024).

Further, we investigate the effect of synthetic negatives. To this effect, we use a mutation-based approach for synthetically introducing errors into Python code while maintaining its **syntactic correctness**, as detailed in Appendix A.8. This method uses Abstract Syntax Tree (AST) manipulation to apply mutations like argument swapping, operator replacement, control flow changes, off-by-one errors, and return value modification. By injecting these errors, it generates valid but behaviorally altered code. We apply this to on-policy positive examples, creating off-policy negatives for model training under the same setup. Observing that the synthetic negatives could even potentially harm the performance, we confirmed the importance of preference training with more natural, and ideally on-policy negative samples.

This highlights our contribution in demonstrating a method empowered by automated test cases and efficient on-policy preference learning. This approach can be easily adopted to scale the collection of test cases, providing a robust supervision signal for model training.

**Importance of Test Case-Based Preference Learning** We experiment with including only non-executable samples as rejected solutions, while using the same set of chosen solutions. As shown in Figure 2, we observe that this is worse than PLUM in most cases. More importantly, it does not always improve the model's performance and may even hurt. This has also been noted in previous studies (Wang et al., 2024b). Although the positive examples used are the same as our oracle-based preference learning, lower-quality negative examples do not necessarily help the model improve due to the additional noise in the preference signal.

**Test Cases Allow By-passing Reward Model Training For Iterative Alignment** We demonstrate in Table 7 that PLUM enables iterative on-policy alignment while providing accurate preference signals without requiring reward model training. Notably, iterative preference learning, facilitated by the online feedback loops generated through the test case collection procedure, outperforms offline

methods on the challenging LeetCode benchmark, using the same data. This finding aligns with results from other domains (Xiong et al., 2024a;b), further confirming the advantages of online learning. This demonstrates the further potential of PLUM in advancing code language models especially in more challenging problems by its support for efficient online policy improvement.

## 4  RELATED WORKS

**Reinforcement Learning and Preference Learning For Reasoning-Related Tasks**   Preference learning algorithms like Direct Preference Optimization (DPO) (Rafailov et al., 2023) and Kahneman & Tversky's Optimization(KTO) (Ethayarajh et al., 2024) are popular for their cost efficiency and training stability. Beyond controlling model-user interactions, these methods are now applied to more complex reasoning tasks. Ocra-Math (Mitra et al., 2024) uses iterative preference learning to improve math reasoning in SFT-ed language models, while Eurus (Yuan et al., 2024) leverages preference trees for solving complex problems through multi-step interactions with external feedback.

**Code Generation with Large Language Models**   Code generation has become a key application of generative language models. Pre-training on code corpora has led to strong performance in models like StarCoder (Li et al., 2023), StarCoder2 (Lozhkov et al., 2024), and DeepSeek-Coder (Guo et al., 2024a), while others like CodeQwen (Bai et al., 2023) and CodeLlama (Rozière et al., 2024) benefit from continued pre-training on additional code data. To enhance these models, supervised fine-tuning on instruction-response pairs has been employed (Luo et al., 2023; Wei et al., 2023; Zheng et al., 2024). Reinforcement learning techniques, like those in CodeRL (Le et al., 2022; Shojaee et al., 2023; **?**), and reward models used in DeepSeek-Coder-V2 (DeepSeek-AI et al., 2024) also improve performance using test feedback.

**Test Case Generation with Language Models**   Automated test case generation (Pacheco et al., 2007; Fraser & Arcuri, 2011; Panichella et al., 2015) is crucial for ensuring software quality and safety and has long been a key topic in software engineering. The advent of LLMs has inspired works using transformer models for test generation, either by training models (Tufano et al., 2021; Li et al., 2022) or prompting them (Chen et al., 2023). Test cases also help clarify user intent, aligning model-generated programs with user requirements (Fakhoury et al., 2024; Endres et al., 2024).

**The synergy between test cases and code generation**   Programming-by-examples (Gulwani, 2016) and test-driven programming (Perelman et al., 2014) focus on using test cases to automatically refine programs to meet specifications. This concept has been adapted to enhance deep learning approaches to code synthesis (Kulal et al., 2019; Chen et al., 2023; Zelikman et al., 2023). Recent methods, like CodeT (Chen et al., 2023) and Parsel (Zelikman et al., 2023), use test cases to reduce the search space during inference, while ALGO (Zhang et al., 2023) employs brute-force solutions as oracles to generate test outputs for competitive programming. Our approach, similar to Haluptzok et al. (2023), leverages test cases during training to improve models' inherent programming capabilities.

## 5  CONCLUSION

In this paper, we introduced PLUM, a novel preference learning framework designed to improve the ability of code language models (LMs) to distinguish between correct and incorrect code by leveraging test cases. Our framework tackles the limitations of traditional supervised fine-tuning approaches by embedding on-policy learning directly into the training process. Through the automatic generation and evaluation of test cases, PLUM enables models to learn from their own outputs without requiring separate reward models or manual labeling, offering a scalable and flexible solution.

The results from our experiments demonstrate the effectiveness and generalizability of PLUM. Furthermore, we performed careful experiments and showed that on-policy preference learning outperforms various off-policy methods, highlighting the crucial role played by on-policy training. Further, we demonstrated PLUM allows for effective online preference learning that further pushes the performance on challenging coding benchmarks.

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

# A APPENDIX

## A.1 TEST CASE GENERATION

To produce test cases for each programming instruction, we queried OpenAI GPT-4 with temperature 0 and max response token 4096.

## A.2 TRAINING DETAILS

We trained the model using the KTO objective, with a learning rate $5 \times 10^{-7}$, linear scheduler, $\beta = 0.1$, and maintained the desirable-to-undesirable ratio to be 1. We train each model using 8-bit quantized LoRA for 3 epochs on a Nvidia A-100 GPU with 40 GB memory.

## A.3 ADDITIONAL RESULTS: WHAT IF DIRECTLY USING COMPETITIVE CODING DATASETS?

There are existing competitive coding datasets like CODE CONTESTS Li et al. (2022) which are equipped with test cases. Using more general datasets like OSS-INSTRUCT (Wei et al., 2023), paired with synthetic test cases, is more advantageous for preference learning in code language models than using competitive coding datasets like Code Contests. Competitive coding datasets present complex problems with intricate edge cases, which can overwhelm the model and obscure fundamental instruction-following and preference-learning goals. In contrast, OSS-Instruct provides more accessible, more uniform instructions that allow for cleaner and more straightforward alignment. This helps models learn functional correctness more effectively without being distracted by the nuances of competitive coding. Additionally, OSS-Instruct, sourced from real-world open-source projects, avoids domain-specific biases that can arise from competitive coding, making it more generalizable and applicable across diverse programming environments.

We also conducted experiments to repeat the same process using CODECONTESTS dataset. Due to the challenging nature of this dataset, we sampled more to get the same number of positive and negative cases as in OSS-INSTRUCT and SHAREGPT etc. As shown in Table 8, training on this dataset seems ineffective.

|  | Item | MBPP | MBPP+ | HE | HE+ | Avg. |
|---|---|---|---|---|---|---|
| MAGICODER-S-DS | Baseline | 75.7 | 64.4 | 76.8 | 70.7 | 71.9 |
|  | KTO | 74.9 | 64.7 | 75.4 | 72.6 | 71.9 |
|  | DPO | 74.9 | 64.9 | 76.8 | 71.3 | 72.0 |
| MAGICODER-DS | Baseline | 75.4 | 61.9 | 66.5 | 60.4 | 66.1 |
|  | KTO | 75.7 | 63.2 | 65.2 | 59.8 | 66.0 |
|  | DPO | 75.7 | 63.2 | 65.2 | 59.8 | 66.0 |

Table 8: Training on Code Contests

## A.4 RESULTS ON BASE MODELS

Below we present the results of directly applying PLUM on base models without performing supervised fine-tuning and the comparison with training using SFT in Tables 9 and 10.

| Model | Type | MBPP | MBPP+ | HE | HE+ | Avg. (Base) | Avg. (+) | Avg. (All) |
|---|---|---|---|---|---|---|---|---|
| STARCODER2-BASE | Baseline | 54.4 | 45.6 | 35.4 | 29.9 | 44.9 | 37.8 | 41.3 |
| | SFT | 62.2 | 49.4 | 41.5 | 35.4 | 51.9 | 50.6 | 47.1 |
| | **PLUM-KTO** | 60.4 | 49.1 | 46.3 | 39.6 | **53.4** | **51.2** | **62.2** |
| CODEQWEN-BASE | Baseline | 72.2 | 60.2 | 51.8 | 45.7 | 62.0 | 53.0 | 57.5 |
| | SFT | 73.4 | 62.4 | 67.7 | 59.1 | 70.6 | 66.5 | 65.7 |
| | **PLUM-KTO** | 75.4 | 62.9 | 70.1 | 62.2 | **72.8** | **67.8** | **67.7** |
| DS-CODER-BASE | Baseline | 70.2 | 56.6 | 47.6 | 39.6 | 58.9 | 57.8 | 53.5 |
| | SFT | 71.7 | 57.1 | 56.1 | 48.8 | 63.9 | 60.5 | 58.4 |
| | **PLUM-KTO** | 72.9 | 58.9 | 56.7 | 48.8 | **64.8** | **61.9** | **59.3** |

Table 10: PLUM vs. SFT for base models.

| Model Families | Item | MBPP | MBPP+ | HE | HE+ |
|---|---|---|---|---|---|
| CODEQWEN-BASE | Base | 72.2 | 60.2 | 51.8 | 45.7 |
| | **OSS-Instruct** | **75.4** | **62.9** | **70.1** | **62.2** |
| | Rel. + | 4.4 | 4.5 | 35.3 | 36.1 |
| | **ShareGPT-Python** | **76.4** | **64.9** | **73.2** | **67.1** |
| | *Rel. +* | *5.8* | *7.8* | *41.3* | *46.8* |
| DS-CODER-BASE | Base | 70.2 | 56.6 | 47.6 | 39.6 |
| | OSS-Instruct | 72.9 | 58.9 | 56.7 | 48.8 |
| | Rel. + | 3.9 | 4.1 | 19.1 | 23.2 |
| | ShareGPT-Python | 75.4 | 60.7 | 64 | 53.7 |
| | Rel. + | 6.4 | 7.2 | 34.5 | 35.6 |
| STARCODER2-BASE | Base | 54.4 | 45.6 | 35.4 | 29.9 |
| | **OSS-Instruct** | **60.4** | **49.1** | **46.3** | **39.6** |
| | Rel. + | 11 | 7.7 | 30.8 | 32.4 |
| | **ShareGPT-Python** | **63.9** | **51.9** | **50** | **42.1** |
| | Rel. + | 17.5 | 13.8 | 41.2 | 40.8 |

Table 9: Results on base model training.

## A.5 DISTRIBUTION OF POLICY MODEL CORRECTNESS

Figure 3 shows the pass ratio on OSS-Instruct dataset of models we consider in this study.

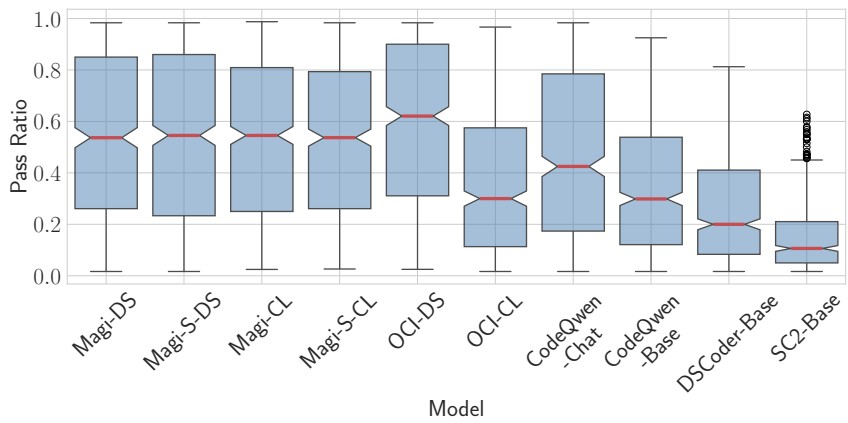

Figure 3: Distribution of policy model correctness ratio on OSS-Instruct dataset.

| Test Generator | Algorithm | MBPP | MBPP+ | HE | HE+ | Avg. | LeetCode | LCB |
|---|---|---|---|---|---|---|---|---|
| - | Baseine | 77.7 | 67.2 | 83.5 | 78.7 | 76.8 | 33.9 | 23.2 |
| GPT-4 | KTO | 81.0 | 69.0 | 86.0 | 81.1 | 79.3 | 35.2 | 25.8 |
| | DPO | 81.2 | 70.2 | 86.0 | 81.1 | 79.6 | 36.7 | 25.8 |
| LLAMA3-70B | KTO | 79.4 | 69.9 | 84.8 | 80 | 78.5 | 36.7 | 24.5 |
| | DPO | 79.4 | 66.2 | 84.1 | 79.3 | 77.3 | 36.1 | 24.5 |
| LLAMA3-405B | KTO | 79.4 | 67.7 | 84.1 | 79.9 | 77.8 | 36.1 | 23.8 |
| | DPO | 79.2 | 66.9 | 85.4 | 80.5 | 78.0 | 36.6 | 25.5 |
| GPT-3.5-TURBO | KTO | 80.2 | 67.9 | 84.8 | 79.9 | 78.2 | 36.1 | 24.0 |
| | DPO | 79.7 | 67.7 | 84.8 | 79.9 | 78.0 | 36.1 | 23.8 |
| GPT4O-MINI | KTO | 81.2 | 69.2 | 85.4 | 80.5 | 79.1 | 36.7 | 24.0 |
| | DPO | 80.5 | 67.6 | 85.4 | 81.1 | 78.7 | 36.7 | 23.8 |
| CLAUDE-3-HAIKU | KTO | 79.7 | 67.2 | 85.4 | 81.7 | 78.5 | 36 | 24.0 |
| | DPO | 79.9 | 67.2 | 86 | 81.7 | 78.7 | 36 | 25.5 |

Table 11: Ablation of test case generator models. We used CODEQWEN-1.5-CHAT as the policy model. LCB stands for LiveCodeBench.

| Model | Item | LeetCode | LiveCodeBench |
|---|---|---|---|
| QWEN-2.5-INSTRUCT-14B | **Baseline** | 55.0 | 46.0 |
| | **PLUM-DPO** | **58.3** | **47.0** |
| QWEN-2.5-CODER-14B | **Baseline** | 58.3 | 32.2 |
| | **PLUM-DPO** | **61.7** | **35.0** |

Table 12: PLUM on more powerful policy models.

## A.6   ABLATION ON TEST CASE GENERATOR

To demonstrate the robustness of PLUM across different models as test case generators, and more specifically, to showcase its ability to boost the policy model's performance with more efficient options for test case generator thus proving its scalability, we conduct extensive experiments with multiple other test case generator models. We considered proprietary models with much more affordable API access and presumably less powerful than GPT-4 (GPT-3.5-Turbo, GPT4o-mini, Claude-3-Haiku), and Open-weight models (Llama 3.1-70B and 405B).

Table 11 displays the results of the ablation study. Consistent performance gains are observed across the experiments.

Importantly, the use of more cost-efficient test case generators does not compromise PLUM's effectiveness. This demonstrates the scalability of our approach, enabling its practical application across a wide range of test generators and resource constraints.

## A.7   IMPROVING STRONGER LANGUAGE MODELS

To validate the generalizability of the approach to more powerful post-trained models with larger parameter size and more sophisticated training, we conducted experiments with QWEN-2.5-INSTRUCT-14B (Yang et al., 2024a) and QWEN-2.5-CODER-14B (Hui et al., 2024) models. These models are fine-tuned from larger and stronger pre-trained models have undergone more sophisticated post-training including reinforcement learning and preference alignment.

We maintain the same setting as OSS-Instruct experiments in Section 3.2. As presented in Table 12, PLUM can further improve these models' performance. This not only validates the effectiveness of our approach, but highlights the potential of PLUM to be applied to complement other post-training techniques.

## A.8   GENERATION OF SYNTHETIC NEGATIVES

We present the algorithm we used to generate synthetic negatives below in Algorithm A.8.

**Algorithm 2** MutateCode Algorithm

**Require:** *source_code* as a string, mutation probability $P$
**Ensure:** *mutated_code* as a string
 1: **Parse** the source code into an AST: *tree* ← ParseAST(*source_code*)
 2: **Initialize** mutation rules:
 3:     Swap function arguments
 4:     Change arithmetic/logical operators
 5:     Modify control flow (negate conditions, swap if-else blocks)
 6:     Introduce off-by-one errors in loops
 7:     Remove exception handling blocks
 8:     Alter return values
 9: **Define** *Mutator* class:
10:     **Function** visit_FunctionDef(node):
11:         Store function signatures, recursively traverse AST
12:     **Function** visit_Assign(node):
13:         Track variable types, recursively traverse AST
14:     **Function** visit_Call(node):
15:         With probability $P$, swap arguments if types match
16:         With probability $P$, replace function call with another compatible one
17:     **Function** visit_If(node):
18:         With probability $P$, negate the condition or swap if-else blocks
19:     **Function** visit_For(node):
20:         With probability $P$, introduce off-by-one error in loop range
21:     **Function** visit_Try(node):
22:         With probability $P$, remove exception handling block
23:     **Function** visit_Return(node):
24:         With probability $P$, alter the return value
25: **Apply** the Mutator to the AST: *mutated_tree* ← Mutator().visit(*tree*)
26: **Perform** syntactic validation: *is_valid* ← SyntaxCheck(*mutated_tree*)
27: **if** *is_valid* = **False then**
28:     **return** original source code or error
29: **end if**
30: **Convert** the mutated AST back to code: *mutated_code* ← ASTtoSource(*mutated_tree*)
31: **return** *mutated_code*

