# OpenReview forum: "$\textbf{PLUM}$: Improving Code LMs Using On-Policy Preference Learning Powered by Automatic Test Cases"
_ICLR.cc/2025/Conference — ICLR 2025 Conference Withdrawn Submission_

### Official Review · Reviewer_AYWK · 2024-10-30

**Soundness:** 3
**Presentation:** 2
**Contribution:** 3
**Rating:** 6
**Confidence:** 3

**Summary:**

The paper proposes a preference learning framework (PLUM) for automatically building code preference data, which uses LLMs with natural language instructions to incorporate test cases and the model's on-policy candidate solutions into the training process. On commonly used evaluation benchmarks: HumanEval(+) and MBPP(+), PLUM further pushes performance of a wide range of instruction models on these coding benchmarks.

**Strengths:**

1. Without RM, this method accurately captures preferences related to accuracy with test cases and effectively improves the code capability of the model.
2. This method exhibits good generalization capabilities, as it can integrate with other preference methods to further enhance the performance of various code instruction models.

**Weaknesses:**

1. The paper uses GPT-4-1106 as the generator model to obtain test cases. Without a more powerful model, can this method still have a significant improvement? Can you conduct an ablation study using less powerful models (e.g. GPT-3.5) for test case generation to analyze how model capability affects the overall performance gains？
2. There is no experimental verification for the enhancement capability of this method towards more powerful models, such as DeepSeek-Coder-V2-Instruct(21B AP / 236B TP). Can you add some experiments based on stronger models？

**Questions:**

1. The pseudocode is inconsistent with the description in the paper. The paper talks about " Solutions passing all test cases are used as the chosen solutions, and those failing at least one the rejected solutions". But in pseudocode line 14, It seems like having just one case makes it a positive instance.

---

> ### Author Response · Authors · 2024-11-28
>
> We sincerely thank the reviewer for recognizing our contributions and providing detailed feedback that helps refine our work. Below, we address your insightful comments and suggestions.
>
> ### Ablation Shows PLUM Works For Various (Especially Less-Powerful) Test Generators
>
> We conducted an ablation study using alternative test case generators to evaluate PLUM's scalability and cost-efficiency.
> The study included open-weight models like Llama-3.1-70B and cost-effective API-based models such as GPT-3.5-Turbo, along with the lowest-cost models from OpenAI (GPT4o-mini) and Anthropic (Claude3-Haiku). The results consistently show performance improvements, even with less powerful test case generators, demonstrating PLUM's robustness to generator variability and its applicability under different resource constraints.
> For detailed results, please refer to the global response and our revised draft.
>
>
> ### Proving PLUM’s Effectiveness on Stronger Models
>
> We evaluated PLUM on stronger models:  Qwen-2.5-Instruct-14B and Qwen-2.5-Coder-14B. These models already exhibit strong performance on coding benchmarks, and have undergone more sophisticated alignment techniques.
>
> Our results show that applying PLUM further improves these models. This proves that PLUM can enhance the performance of stronger models and highlights its complementary effect when combined with other alignment techniques. The results have been also included in the revision.
>
> | **Model**             | **Item**  | **LeetCode** | **LiveCodeBench** |
> |-----------------------|-----------|:------------:|:-----------------:|
> | Qwen-2.5-Instruct-14B | Baseline  |     55.0     |        46.0       |
> |                       | PLUM-DPO |     **58.3**     |        **47.0**       |
> | Qwen-2.5-Coder-14B    | Baseline  |     58.3     |        32.2       |
> |                       | PLUM-DPO |     **61.7**     |        **35.0**       |
>
> ### Pseudocode Typo
> Thanks for pointing out the mistake! Your understanding is correct that “Solutions passing all test cases are used as the chosen solutions, and those failing at least one are rejected solutions.”  We have fixed this in the revision.

---

> > ### Comment · Reviewer_AYWK · 2024-11-28
> >
> > Thanks for the effort. The response resolves my concern.

---

> > > ### Author Response · Authors · 2024-11-28
> > >
> > > Thank you for your prompt response! We would like to thank you again for the valuable comments in shaping our work!

---

> > > > ### Author Response · Authors · 2024-12-03
> > > >
> > > > Dear Reviewer AYWK,
> > > >
> > > > We thank you again for your positive feedback and constructive comments for us to further improve our work!
> > > >
> > > > We would like to see if you have any further questions or concerns regarding this work that we could clarify further!
> > > >
> > > > We also hope the improvements we’ve made will resonate positively with your evaluation. Your perspective plays a crucial role in shaping the paper’s chances to acceptance, and we truly value your thoughtful assessment!
> > > >
> > > > Thank you very much again for your efforts!

---

### Official Review · Reviewer_6RyX · 2024-11-03

**Soundness:** 3
**Presentation:** 3
**Contribution:** 3
**Rating:** 8
**Confidence:** 4

**Summary:**

The current research piece introduced a new approach for improving code generation skills on existing Code LLMs. To this end, a two step process is introduced i) given a set of coding problems, generate, via prompting, test cases for them, and validate their correctness with the reference solution ii) with the resulting dataset, prompt the LLM to generate new solutions, check them with the generated test cases, and updated the model via iterative preference optimization.
The idea is applied to several standard Code LLM models using DPO and KTO as preference optimization technique and compared with other, non PO approaches including prompting and alternative fine-tunning techniques.

**Strengths:**

The work is reasonable and clear. It uses established preference optimization techniques as DPO and KTO on for code synthesis improvement. It relies on automatic generation of test cases, guaranteeing their correctness by checking them with the reference solution.
Experimentation show a clear gain across the board in all the models included in the experiments (which are the most popular ones) to avoid the report of any spurious case.

**Weaknesses:**

The work is solid.
The novelty aspect might be the only low point compared to the rest of the work. All the individual aspects on the technique: generating test cases, filtering them, using that as corpus, iterative PO, are all steps previously used on the improvement of Code LLMs

**Questions:**

Some suggestions

* Tables:
Table 3 is shown after table 4. Tables numbers should follow their occurrence order.
Also, table 2, seems too far from where is referenced (shown in page 5, referenced at the bottom of page 7.
Lines 350 - 364 encloses Table 3 despite it refers to Table 2.

**Details Of Ethics Concerns:**

--

---

### Official Review · Reviewer_Du4Y · 2024-11-04

**Soundness:** 3
**Presentation:** 3
**Contribution:** 2
**Rating:** 3
**Confidence:** 4

**Summary:**

The paper presents an on-policy preference learning framework augmented with test cases for code language models called PLUM. It enhances the code generation capabilities of language models by leveraging preference learning driven by automatically generated test cases. The preference data is curated by evaluating candidate code solutions sampled from the policy based on the test cases.  PLUM is evaluated on several code generation tasks, including HumanEval, MBPP and LiveCodeBench, and shows improvements over SFT’ed models and other execution-feedback-driven methods.

**Strengths:**

- The proposed method uses on-policy preference learning, which aligns with the distribution of the base model itself, reducing the risk of distribution shift.

- The authors conduct comprehensive experiments with several models and compare their performance on multiple benchmarks.

- PLUM does not require extra training for the reward models, which simplifies its implementation.

**Weaknesses:**

- This paper focuses on Python language only, during training and evaluation. It would be better if the authors could discuss how the methods could be applied to multilingual settings. Since different programming languages have different executing requirements, I assume it’s not very straightforward to apply the proposed framework directly to other languages.

- The preference learning relies on executing the test cases, which need to be generated by GPT-4, which is not scalable to generate large volumes of data, limiting the preference data size. Exploring other open-source LLMs might be more helpful in understanding the robustness of the proposed method, i.e. whether it heavily relies on the power of GPT-4.


- The model sizes used in the experiments are not clearly explained. There are several model families that have different sizes of instruct-model, such as DeepSeek-Coder-6.7B-Instruct and DeepSeek-Coder-33B-Instruct. Without such information, it is harder to understand the proposed method’s impacts on different model sizes.

**Questions:**

When checking the consistency between the generated reference solution and the test cases, in line 179,  how to check whether the test cases accurately reflect the solution?

---

> ### Author Response · Authors · 2024-11-17
>
> Thank you for the insightful feedback, which has helped clarify key aspects of our work.
> # Limitation to Python
> We appreciate the reviewer’s comment on the scope of our work, which currently focuses on the Python programming language. The proposed PLUM framework, however, is designed to be broadly applicable across programming languages. Our approach utilizes test cases generated based on natural language instructions and verifies code through test execution rather than relying on any language-specific syntax or semantics unique to Python. The focus on Python was primarily for the availability of well-established training datasets and ease of comparison and reproducibility, as Python is widely used in both code generation and evaluation benchmarks (e.g. HumanEval, LeetCode). We are confident that the methods can be adapted to multilingual settings by training models on diverse programming languages and modifying test generators to meet language-specific requirements.
> # Dependency on GPT-4
> Thank you for this insightful feedback. Our use of GPT-4 as a generator model was based on its superior generation capabilities; however, our framework does not rely on any features specific to GPT-4. The model was chosen purely for practical reasons, and any other language model capable of generating syntactically correct and logically coherent code solutions would also be effective within our framework. Open-source models can replace GPT-4 for test case generation, allowing for scalable expansion while retaining the effectiveness of our approach. Our experiments verify that the robustness of our method stems from the structured preference learning pipeline rather than reliance on GPT-4’s unique characteristics.
> # Clarification of Model Sizes
> We appreciate the constructive feedback on the clarity of the paper. To clarify the reviewer’s concern regarding model sizes, we aim to illustrate that our ablation studies leverage responses from models of varying sizes specifically to demonstrate the comparative benefits of on-policy data. We did not actually train models of multiple sizes, but rather utilized the outputs from both smaller and larger models as part of our on-policy versus off-policy comparisons. This setup is intended to evaluate and emphasize the effectiveness of our preference learning method across different model capacities without requiring separate training instances.
> Specifically, the models we trained are all 7B-scale language models. CODEQWEN-1.5-CHAT refers to the 7B CodeQwen-1.5-Chat model, Magicoder-DS (-S-DS) refers to the Magicoder models trained from DeepSeek-Coder-6.7B base model, Magicoder-CL (-S-CL) refers to the Magicoder models trained from CodeLlama2-7B base model. Same applies to OCI (OpenCodeIntepreter). We will update the paper to make it clear!
> # Consistency of Test Cases and Reference Solutions
> We appreciate the reviewer’s inquiry into test case consistency with reference solutions. In our approach, consistency between test cases and reference solutions is crucial, but as highlighted in works on software testing, achieving perfect alignment between natural language specifications and test cases remains an open research question. We adopt a practical heuristic by assuming that consistency between generated solutions and passing test cases implies correctness. Our experimental results demonstrate that this assumption holds well across our chosen datasets, which suggests that it is a reasonable approach under current constraints.
>
> We hope that addressing these points will provide additional insights into the PLUM framework, potentially enhancing the reviewers' understanding and appreciation of our contributions and reflecting positively in their assessment.

---

> ### Author Response · Authors · 2024-11-23
>
> Dear Reviewer Du4Y,
>
> Thank you for your valuable feedback on our submission! We have responded your comments in our rebuttal and would appreciate any further clarifications or discussions to enhance our work!
>
> Look forward to further discussions!
>
> Thanks again for your time and efforts.

---

> > ### Comment · Reviewer_Du4Y · 2024-11-24
> > **Response to authors**
> >
> > Thanks for addressing the points raised in weakness. Please find my feedback below.
> > ## On language expansion:
> > While I appreciate your explanation, the process of generating test cases that meet language-specific requirements involves non-trivial efforts, including setting up appropriate executing environments. Without a clearer estimation of these efforts and their feasibility, the generalizability of your approach remains unclear to me.
> > ## On reliance on GPT-4:
> > My concerns about dependency on GPT-4 have not been fully addressed. As you stated, “Our use of GPT-4 as a generator model was based on its superior generation capabilities,” which indicates a significant reliance on this generator model's performance. Although you mention that “the robustness of our method stems from the structured preference learning pipeline rather than reliance on GPT-4’s unique characteristics,” it would be helpful if you could elaborate on which specific experiments support this claim.

---

> > > ### Author Response · Authors · 2024-11-30
> > > **Follow-up with reviewer Du4Y**
> > >
> > > Dear Reviewer Du4Y,
> > >
> > > Thank you for your time and effort in reviewing our submission. We greatly appreciate your detailed feedback, which helped me identify and address the concerns you raised.
> > >
> > > We have carefully addressed the concerns and responded to your earlier questions in both the previous response and the global comment, supplemented by additional experimental results. We wanted to kindly follow up to ensure we’ve adequately clarified your points and to ask if there is anything further we could elaborate on or improve.
> > >
> > > Thank you again for your invaluable feedback and for your contributions to the review process!

---

> ### Author Response · Authors · 2024-11-28
>
> ### Language expansion
> We agree that Python's ease of environment setup significantly contributes to its popularity in execution-based evaluation. Many recent papers like SelfCodeAlign (NeurIPS’24) [1] and more recent recent RLEF from MetaAI [2] also focus solely on python.
> That said, setting up test environments especially for function-level code is also straightforward, one could build execution environments like MultiPL-E [3] did!
>
> For example, if one wants to test java functions, they could simply install JDK, and prepare the file as shown below:
> ```
> public class AddTest {
>     public static int add(int a, int b) {
>         return a + b;
>     }
>
>     public static void main(String[] args) {
>         assert add(2, 3) == 5 : "Test failed";
>         assert add(0, 0) == 0 : "Test failed";
>         System.out.println("All tests passed");
>     }
> }
> ```
> And run
> ```
> >>> javac AddTest.java
> >>> java AddTest
> ```
>
> To get the results!
>
> **References**
>
> [1] Wei et al. 2024. SelfCodeAlign: Self-Alignment for Code Generation. https://arxiv.org/abs/2410.24198.
>
> [2] Gehring et al. 2024. RLEF: Grounding Code LLMs in Execution Feedback with Reinforcement Learning. https://arxiv.org/abs/2410.02089
>
> [3] Cassano et al. 2022. MultiPL-E: A Scalable and Extensible Approach to Benchmarking Neural Code Generation
> https://arxiv.org/abs/2208.08227
>
> **GitHub**: https://github.com/nuprl/MultiPL-E
>
> ---
>
> ### Reliance on GPT-4
>
> We conducted additional experiments to demonstrate the effectiveness of our framework when using other language models (both open-source and proprietary) as test case generators. We kindly direct the reviewer to our global response for detailed results.
>
> We experimented with various test case generators, including both proprietary models with much more affordable API access and **presumably less powerful than GPT-4** (GPT-3.5-Turbo, GPT4o-mini, Claude-3-Haiku), and **open-weight models** (Llama 3.1 70B and 405B).
>
> We have observed consistent performance gains across the experiments. This proves that **PLUM does not rely on a single powerful model as its underlying test generator**, but is a robust, adaptable and scalable framework that works well under various conditions. In particular, it works under **more cost-and-time efficient set-ups**, making it a scalable to large data volumes.

---

> > ### Author Response · Authors · 2024-12-02
> >
> > Dear Reviewer Du4Y,
> >
> > Thank you for your time and effort in reviewing our submission. We sincerely appreciate your detailed and thoughtful feedback, which has been valuable in helping us identify and address the concerns you raised!
> >
> > In light of the approaching end of discussion period, we would kindly check with you again whether our earlier response could address your concern, and if you have additional comments.

---

> > > ### Author Response · Authors · 2024-12-04
> > >
> > > Dear Reviewer Du4Y,
> > >
> > > Thank you for your thoughtful feedback! We have worked hard to address your concerns and would greatly appreciate it if you could review our response before the rebuttal period ends!
> > >  If you feel our revisions and responses have resolved your points, we’d be truly grateful if you could consider updating your score.
> > >
> > > Thank you for your time and support!

---

### Official Review · Reviewer_RrMw · 2024-11-04

**Soundness:** 2
**Presentation:** 2
**Contribution:** 2
**Rating:** 5
**Confidence:** 4

**Summary:**

This paper explores the application of preference optimization on code language models. To obtain the preference data over pre-defined prompts, the paper follows previous work and uses GPT-4 to automatically generate unit tests (filtered with consistency checks) which are subsequently used to determine whether on-policy code solutions are functionally correct (i.e. preferred) or not (i.e. dis-preferred).

The experiments focus on applying the DPO and KTO methods over these derived data and comparing against SFT and RL (limited to Reflexion) training across multiple models and datasets. The results show consistent improvement of preference optimization over the baselines.

**Strengths:**

- The experiments demonstrate that the proposed PLUM methodology leads to consistent improvement over baselines across many models and datasets.
- The experiments are robust, covering many models and benchmarks.

**Weaknesses:**

- The novelty of the proposed work seems limited to the application of preference optimization over code. The on-policy data are obtained following previous work and the benefits of on-policy vs. off-policy training and test-cased filtered data vs. execution filtered data has already been demonstrated in previous work as well (i.e. Le et al. 2022, Gorinski et al. 2023, Liu et al. 2023).
- The motivation of the paper is somewhat unclear. Preference optimization is generally used to align LLM output to human values, e.g. reduce harmful and toxic expressions. In terms of code solutions that are strictly correct or incorrect (based on functional execution over unit tests), RL or contrastive learning over online signal seem more appropriate, and already explored in previous work. The paper should directly compare against those methods (e.g. CodeRL, DeepSeek-Coder-V2, etc.) instead of Reflexion.
- The presentation of the paper could be improved, with the tables being ordered consistently with how the corresponding experiments are introduced. Corresponding tables and figures should also be moved closer to where they are first referenced.

**Questions:**

- Preference optimization over code has been explored in previous work (Miao et al. 2024, Zhang et al. 2024, Gee et al. 2024). While this work can be considered concurrent, the paper would benefit from discussing it.
- The methodology seems to require a more powerful model (than the one being trained) to generate the unit tests. Have the authors explored using the policy itself to generate the unit tests?
- Performing SFT over OSS-Instruct, Evol-Instruct-Code and ShareGPT may not necessarily translate to improvements in MBPP and HumanEval datasets. SFT may be forcing models to adapt to out-of-domain prompts resulting in lower performance (and an unfair baseline). The benefits observed in PLUM could be due to these datasets being filtered through on-policy consistency.
- Many tables are underexplained and potentially misleading (the best results are not always bolded, as in Table 2, 5 and 7). Please clearly indicate which baseline is used for each experiment. Figure 2 is particularly unclear, and (if this reviewer's interpretation is correct) in many cases PLUM introduces no benefit and in some execution based signal outperforms PLUM. Please confirm.

---

> ### Author Response · Authors · 2024-11-17
> **Response to Reviewer RrMw (1 of 2)**
>
> # Motivation
> We appreciate your insightful comments on the motivations for preference learning within the coding domain. While preference learning has traditionally been applied in domains such as human alignment and detoxification, recent work has demonstrated its potential in more complex, outcome-driven tasks across areas like reasoning, mathematics, and coding, where achieving functional correctness is paramount (Dong et al., 2023; Pang et al., 2024; Yuan et al., 2024; Xiong et al., 2024; Lai et al., 2024; Xie et al., 2024; Weyssow et al., 2024).
>
> In the context of PLUM, we focus on encoding functional correctness directly within the preference objective, which offers a new perspective. Our method leverages test-based signals to ensure correctness through a preference-guided approach, thus removing the need for complex reward model setups while addressing potential over-optimization and distributional shift issues. Additionally, our comparative results, including contrastive learning baselines, underscore PLUM's efficacy in reliably improving functional correctness, as it harnesses the advantages of preference learning to a degree not yet fully explored in the code generation domain.
>
> # Comparison to DeepSeek-Coder-V2 and CodeRL
>
> CodeRL and PLUM represent fundamentally different methodologies for improving code generation. CodeRL is an iterative code-refinement framework build upon reinforcement learning principles. It evaluates and iteratively adjusts failed solutions with a trained critic network. This involves continuous refinement, leveraging test cases during inference to repeatedly improve solutions until they pass.
>
> In contrast, PLUM operates as an on-policy preference learning framework designed to improve a model's intrinsic ability to generate correct code based on natural language prompts, bypassing the need for external critics or iterative refinement. By leveraging automatically generated test cases as a lightweight, consistent feedback mechanism for training, PLUM enables on-policy training without reliance on offline data, mitigating the distributional shift commonly seen in offline models​.
>
> Finally, the works cited by the reviewer and within our paper neither diminish nor overlap with PLUM’s unique contribution, as they are concurrent or fundamentally distinct. PLUM leverages on-policy preference learning with test cases to directly enhance model performance during training using preference learning algorithms. Our framework is distinct in its simplicity and robustness, bypassing reward models and reinforcement feedback entirely. This approach, combined with automated test case generation for scalable, policy-consistent preference data, establishes PLUM as an original solution that fills an unaddressed gap in nl-to-code generation.
>
> # Clarifications on the SFT Baseline
>
> To avoid possible confusion, we clarify that the baseline rows we refer to as “Baseline” in the result tables represent existing models that have been fine-tuned on their respective datasets, not models we further fine-tuned on other data. Therefore, no additional out-of-domain adaptation effects have been introduced.
> In the rejection-sampling fine-tuning (RFT) baseline, we filtered out responses that did not pass initial test cases, leaving only correct responses. The model is then SFTed on the same set of questions as PLUM paired with these positive responses. By comparing PLUM to RFT, we demonstrate that the performance gains of PLUM are not merely due to filtering out those “out-of-domain” questions, but through learning to distinguish between correct and incorrect responses through on-policy training.
> We would also like to emphasize that PLUM is designed as an enhancement layered on top of the supervised fine-tuning (SFT) stage. The SFT models used in our study were already extensively fine-tuned on large and diverse instruction datasets. PLUM adds a further improvement step, where the preference learning framework, augmented with test cases, fine-tunes the model to prioritize functionally correct responses more effectively.
>
> # Questions to Reviewers
>
> We appreciate the reviewer’s suggestions and reference pointers. However, we would be grateful if the reviewer could provide the titles and/or links to those papers for further clarification, since all references seem to be missing. (Le et al. 2022, Gorinski et al. 2023, Liu et al. 2023, Miao et al. 2024, Zhang et al. 2024, Gee et al. 2024)

---

> ### Author Response · Authors · 2024-11-17
> **Response to Reviewer RrMw (2 of 2)**
>
> # Readability / Clarity
> Thank you for your suggestion to improve the clarity of the tables and figures. We will reorganize the tables to follow the order of introduction in the main text, ensuring each experiment and its results are intuitively located near its description. Furthermore, for Figure 2, we will include additional annotations to clarify that the purpose of this figure is to contrast non-executable signals against those tested with our on-policy approach. The result actually shows PLUM’s effectiveness in boosting functional accuracy by leveraging test cases, rather than depending solely on signals from compilation failures.
>
> # References:
> [1] Hanze Dong, Wei Xiong, Deepanshu Goyal, Yihan Zhang, Winnie Chow, Rui Pan, Shizhe Diao, Jipeng Zhang, Kashun Shum, and Tong Zhang. Raft: Reward ranked finetuning for generative foundation model alignment.
> URL: https://arxiv.org/abs/2304.06767
>
> [2] Richard Yuanzhe Pang, Weizhe Yuan, Kyunghyun Cho, He He, Sainbayar Sukhbaatar, and Jason Weston. Iterative reasoning preference optimization, 2024.
> URL: https://arxiv.org/abs/2404.19733
>
> [3] Lifan Yuan, Ganqu Cui, Hanbin Wang, Ning Ding, Xingyao Wang, Jia Deng, Boji Shan, Huimin Chen, Ruobing Xie, Yankai Lin, Zhenghao Liu, Bowen Zhou, Hao Peng, Zhiyuan Liu, and Maosong Sun. Advancing llm reasoning generalists with preference trees, 2024.
> URL: https://arxiv.org/abs/2404.02078
>
> [4] Wei Xiong, Chengshuai Shi, Jiaming Shen, Aviv Rosenberg, Zhen Qin, Daniele Calandriello, Misha Khalman, Rishabh Joshi, Bilal Piot, Mohammad Saleh, Chi Jin, Tong Zhang, Tianqi Liu. Building Math Agents with Multi-Turn Iterative Preference Learning, 2024.
> URL: https://arxiv.org/abs/2409.02392
>
> [5] Xin Lai, Zhuotao Tian, Yukang Chen, Senqiao Yang, Xiangru Peng, and Jiaya Jia. Step-DPO: Step-wise Preference Optimization for Long-chain Reasoning of LLMs 2024. URL: https://arxiv.org/abs/2406.18629
>
> [6] Yuxi Xie, Anirudh Goyal, Wenyue Zheng, Min-Yen Kan, Timothy P. Lillicrap, Kenji Kawaguchi, and Michael Shieh. "Monte Carlo Tree Search Boosts Reasoning via Iterative Preference Learning. 2024.
> URL: https://arxiv.org/abs/2405.00451
>
> [7] Martin Weyssow, Aton Kamanda, and Houari Sahraoui. CodeUltraFeedback: An LLM-as-a-Judge Dataset for Aligning Large Language Models to Coding Preferences. 2024.
> URL: https://arxiv.org/abs/2403.09032

---

> ### Author Response · Authors · 2024-11-23
> **Follow-Up on Rebuttal for Clarifications and Discussion**
>
> Dear Reviewer RrMw,
>
> Thank you for your valuable feedback on our submission! We have attempted to respond to your comments in our rebuttal and would appreciate any further clarifications or discussions to enhance our work.
>
> Look forward to further discussions! Thanks again for your time and efforts!

---

> ### Comment · Reviewer_RrMw · 2024-11-23
>
> Given the lack of clarity on the exact citations on my part, I appreciate the difficulty the authors had in addressing some of my concerns. I am clarifying these citations below, could you please clarify the novelty of your unit-test derived signal on the context of this previous work and why these could not constitute baselines for your approach?
>
> Le et al. 2022 -> CodeRL: Mastering Code Generation through Pretrained Models and Deep Reinforcement Learning
>
> Gorinski et al. 2023 -> Automatic Unit Test Data Generation and Actor-Critic Reinforcement Learning for Code Synthesis
>
> Liu et al. 2023 -> RLTF: Reinforcement learning from unit test feedback
>
> Miao et al. 2024 -> Aligning CodeLLMs with Direct Preference Optimization
>
> Zhang et al. 2024 -> CodeDPO: Aligning Code Models with Self Generated and Verified Source Code
>
> Gee et al. 2024 -> Code-Optimise: Self-Generated Preference Data for Correctness and Efficiency

---

> ### Author Response · Authors · 2024-11-26
>
> ## Concurrent Works
> We thank the reviewer for providing the pointers. Some of these works are concurrent to ours and should not negatively impact PLUM’s contribution:
>
> - Miao 2024: Aligning CodeLLMs with Direct Preference Optimization
>
>    https://arxiv.org/pdf/2410.18585
>
>     **Arxiv date: Oct 24, 2024**.
> - Zhang 2024: CodeDPO: Aligning Code Models with Self Generated and Verified Source Code
>
>     https://arxiv.org/abs/2410.05605
>
>     **Arxiv date: Oct 8, 2024**.
> - Gee 2024: Code-Optimise: Self-Generated Preference Data for Correctness and Efficiency
>
>     https://arxiv.org/pdf/2406.12502
>
>    **Arxiv date: Jun 18, 2024**.
>
> ## Differences From The Rest Of Related Works
> The reviewer correctly pointed out the use of test cases for code LMs has been previously explored. The key contribution of this work is a novel on-policy preference learning framework for code LMs, and discuss its differences from previous works below (CodeRL has already been covered in our previous response).
> ### RLTF
> RLTF is an extension of CodeRL. Its key is to use the enumeration of various kinds of errors and carefully handcrafted reward values for training the policy
> Result-wise, the handcrafted RL reward reduces syntax errors and other superficial issues. However, it fails to effectively ensure that the solutions align with NL-specification (Figure 2 of RLTF paper).
>
> In contrast, PLUM excludes syntax errors and other statically invalid programs, and demonstrates that focusing on run-time feedback by execution over test cases could significantly improve code LMs’ NL-to-code capabilities across benchmarks, besides, instead of relying on reward models as in RLTF, PLUM learns directly from the execution results over test cases with algorithms like DPO and KTO.
> ### Automatic Unit Test Data Generation and Actor-Critic Reinforcement Learning for Code Synthesis
>
> They labeled programs with a static-analysis-based test case annotation tool (EvoSuite [1]) to augment the MBPP training split. EvoSuite focuses on ensuring structural coverage—i.e., making every branch of the program runnable—rather than aligning with natural language specifications. The resulting synthetic dataset caused the model to perform worse than the pre-trained checkpoint (as shown in Table 1 of this paper).
>
> In contrast, we developed a scalable method that leverages LLMs to produce high-quality test cases for programming instructions, enabling the collection of on-policy preference data. Leveraging the signal provided by execution against the test cases, our method directly optimizes the model’s functional correctness for NL-to-code tasks. Through extensive experiments, we clearly demonstrate the effectiveness of our approach in improving code LMs.
>
> ### Additional Remarks:
> EvoSuite is limited to strongly-typed languages such as Java, making it incompatible with dynamically-typed languages like Python. Additionally, the extremely low yield rate in Gorinski 2023 —estimated at just 5.57%—renders it impractical for generating test cases. For instance, producing approximately 2,000 examples would require around 360 hours of computation time, assuming one second per test case generation.
> ## Summary
> In summary, PLUM as a novel test-case driven preference learning approach provides fresh insights beyond existing works, showcasing the importance of on-policy preference data. Beyond that, its strong empirical performance further strengthens its value.
>
> [1] EvoSuite: automatic test suite generation for object-oriented software. https://dl.acm.org/doi/10.1145/2025113.2025179

---

> ### Comment · Reviewer_RrMw · 2024-11-28
>
> I would like to thank the authors for providing their insights on the related work, I certainly appreciate their efforts so far. I think the paper would benefit from the inclusion of this discussion, as it places the presented work in the proper context. I agree that much of that work can be considered concurrent and thus should not diminish the paper's contribution (in my original review I suggested that these should be included in the paper for discussion, not comparison).
>
> However, the novelty of the approach is still impacted by the fact that non-concurrent related work has already established that code models can be improved by exploiting on-policy signal assigned a reward through unit tests (either a continuous reward for RL or a binary for PO). At this point it is also important to note that previous work has also established that LLMs can be used to automatically produce unit tests and that their inclusion (either directly in the input or as a filtering step) can improve performance.
>
> Xiong et al 2024 "The Program Testing Ability of Large Language Models for Code"
>
>  Li et al. 2023 "Towards enhancing in-context learning for code generation."
>
> Chen et al. 2023 "CodeT: Code Generation with Generated Tests"
>
> With all that said, to this reviewer, it seems that all individual contributions of the paper (on-policy learning for code, automatic generation of unit tests, filtering of on-policy signal by unit tests) have already been proposed and supported by previous work. Can the authors please help me clarify what is unique to this method, beyond the combination of previously established on-policy methods and signal on code? If the authors claim that nothing is specifically unique, but the proposed variations on these methods perform better, then direct comparisons and ablations are needed.

---

> ### Author Response · Authors · 2024-12-01
>
> We thank the reviewer for their discussions.
>
> > ...non-concurrent related work has already established that code models can be improved by exploiting on-policy signal assigned a reward through unit tests (either a continuous reward for RL or a binary for PO).
>
> We kindly point out that PLUM innovatively leverages test-case-driven preference optimization for code. To the best of our knowledge, **no prior works have demonstrated test-case signals can improve LMs’s functional correctness using a reward-model-free preference learning (xPO) technique**. As the same reviewer **RrMw** correctly pointed out in their earlier comments, _‘preference optimization is (was) generally used to align LLM output to human values, e.g. reduce harmful and toxic expressions’_, it remained elusive before we proposed PLUM whether and how preference learning techniques can go beyond aligning to human preferences and improve the code LMs’ functional correctness when using test case feedback.  PLUM addresses this critical gap and proposes an effective solution, and therefore shall be considered novel and valuable.
>
> Additionally, although _‘RL or contrastive learning over online signal seem more appropriate’_ (quoted from the earlier review by **RrMw**), we demonstrated the strong performance of PLUM, which brings efficient, scalable and effective on-policy improvement for code LMs without the need for reward-models or reward handcrafting.
>
> ---
>
> > At this point it is also important to note that previous work has also established that LLMs can be used to automatically produce unit tests and that their inclusion (either directly in the input or as a filtering step) can improve performance.
>
> These referenced works fundamentally differ from PLUM since they use test cases to rerank / refine solutions during inference time, not using these to establish training signals as PLUM did. Indeed, as we have acknowledged, our test case generation approach partially builds upon the insights that LLMs exhibit the ability to generate high-quality test cases based on its understanding of natural language instructions. However, we clarify that the fundamental difference in the use of these test cases distinguishes PLUM as a **training framework** from the inference time **solution-reranking techniques**.  Therefore, our contribution of proposing a preference learning framework that can build better code LMs from test cases is novel and valuable.
>
> We summarize the works mentioned by the reviewer below.
>
> **Xiong et al. 2024**: Evaluates and improves LLMs' ability to generate test cases for code, achieving pass rate improvements on HumanEval+.
>
> **Li et al. 2023**: Proposes AceCoder, an **in-context** learning method combining test-based analysis and retrieval, significantly enhancing code generation performance.
>
> **Chen et al. 2023**: Introduces CodeT, an **in-context** learning method that generates and uses test cases to improve code correctness, achieving state-of-the-art results on multiple benchmarks.
>
> ---
>
> > all individual contributions of the paper (on-policy learning for code, automatic generation of unit tests, filtering of on-policy signal by unit tests) have already been proposed
>
> The reviewer correctly notes that PLUM builds on prior insights as all research papers do, which we have acknowledged in the paper. But those works typically address a different problem from PLUM.
>
> PLUM uniquely integrates these components into a unified, scalable framework for on-policy preference optimization of code LMs. Unlike prior work, PLUM demonstrates:
> - The effectiveness of on-policy preference optimization for code generation.
> - The utility of LM-synthesized test cases in providing high-quality training signals.
> - The critical importance of on-policy learning, supported by empirical evidence.
>
> Our contributions fundamentally differ from existing techniques, yielding novel insights and performance improvements that prior work neither achieves nor enables.
>
> ---
>
> #### Request To Discuss Concurrent Works
> We appreciate the pointers from the reviewers. In fact, DeepSeek-Coder-V2 has been discussed in the paper as a concurrent work. It was infeasible for us to include works that were public (on Arxiv) after the ICLR deadline (Miao 2024: Aligning CodeLLMs with Direct Preference Optimization, **Arxiv date: Oct 24, 2024**.
> Zhang 2024: CodeDPO: Aligning Code Models with Self Generated and Verified Source Code, **Arxiv date: Oct 8, 2024**) at the time of submission. That said, we are happy to discuss them in future revisions!

---

> ### Author Response · Authors · 2024-12-02
>
> Dear Reviewer RrMw,
>
> Thank you again for the time and effort to review our paper and to initiate discussions that help clarify our contributions!
>
> With 2 days left until the end of the discussion period, it would mean so much to us if you could take a look at the added discussion and materials so that you can re-evaluate our work with those clarifications and update your scores if you find appropriate.
>
> Please let us know any remaining questions / concerns surrounding PLUM's contributions! We are also more than happy to address your further questions and concerns!

---

> ### Author Response · Authors · 2024-12-03
> **RLTF Baseline Results**
>
> As suggested by the reviewer, we applied RLTF on Magicode-DS-6.7B and CodeQwen-1.5-7B-Chat, following their dataset choice (APPS). We appreciate the reviewer's patience, as RLTF (and the precedent RL works) are typically proposed before the prevalence of instruction-tuning for code-LMs!
> ### Results and analysis
> The result is presented below:
> |                      | HumanEval(+) | MBPP(+)      | LeetCode |
> |----------------------|--------------|--------------|----------|
> | CodeQwen-1.5-7B-Chat | 83.5 (78.7)  | 77.7 (67.2)  | 33.9     |
> |     + RLTF                 | 81.7(76.2)  &#8595; | 76.2 (64.4) &#8595; | 33.9   |
> | Magicoder-DS-6.7B    | 66.5(60.4)   | 75.4 (61.9)  | 19.4     |
> |     + RLTF                 | 64.6(59.1) &#8595;  | 75.2 (61.9)  &#8595;| 15.5   &#8595;  |
>
> We notice that RLTF in its original form might not be particularly effective to improve modern instruction-tuned code LMs.
> In addition to the restrictions brought by the training distributions, sparsity of reward could be another reason. Notably, very recent works like Dai (2024) noted on the sparse reward signals inherent in RLTF and demonstrate its limitations in further improving instruction-tuned code language models through experiments.
>
> ---
>
> ### Re-emphasizing difference between PLUM and RLTF
> Again, we emphasize the distinctions between PLUM and RLTF. RLTF focuses on adapting pre-trained code LMs to specific datasets (e.g., APPS) through **reward design**. In contrast, PLUM leverages on-policy preference learning to improve instruction-tuned code LMs by directly leveraging the execution feedback, offering a simpler, more scalable, and effective alternative to RL approaches without the need for RL.
>
> Let us know if you have further questions or concerns that we could take the last chance to clarify within the last day of discussion period!
>
> [1] Dai et al. Process Supervision-Guided Policy Optimization for Code Generation. https://arxiv.org/pdf/2410.17621

---

> > ### Author Response · Authors · 2024-12-04
> >
> > Dear Reviewer RrMw,
> >
> > We sincerely appreciate your valuable feedback, which has helped us refine our work and better situate it within the broader research context. We have made significant efforts to address your questions and have conducted additional baseline experiments that we believe directly respond to your concerns.
> >
> > We hope these improvements bring our work closer to meeting your expectations. We would be deeply grateful if you could consider updating your rating to reflect these revisions and support our submission. We will include the additional discussions and results in our future revisions.
> >
> > Thank you for your time and thoughtful consideration.

---

### Author Response · Authors · 2024-11-28
**[Additional Results] Ablation On Test Case Generators Demonstrates  $\textbf{PLUM}$'s Applicability Across Various Test Generators**

### Ablation On Test Case Generators Demonstrates  $\textbf{PLUM}$'s Applicability Across Various Test Generators
We thank Reviewers **AYWK** and **Du4Y** for their discussion on PLUM's applicability across diverse models as test case generators. To address this, we conducted ablations with
- proprietary models with much more affordable API access and presumably less powerful than GPT-4 (GPT-3.5-Turbo, GPT4o-mini, Claude-3-Haiku). and
- Open-weight models (Llama 3.1 70B and 405B).

Consistent performance gains were observed across the experiments.

Importantly, the use of more cost-efficient test case generators does not compromise PLUM’s effectiveness. This demonstrates the scalability of our approach, enabling its practical application across a wide range of test generators and resource constraints.

Shown below is the result of ablations using CodeQwen-1.5-7B-Chat as the policy model to apply PLUM. We will update this result in our revised draft.

**Remark:** Claude-3-Haiku from Anthropic and GPT4o-mini from OpenAI are currently the most affordable (per-token API cost) and fastest models available from their respective providers.

| **Test Generator** | **Algo** | **MBPP** | **MBPP+** | **HE** | **HE+** | **Avg.** | **LeetCode** | **LiveCodeBench** |
|--------------------|:--------:|:--------:|:---------:|:------:|:-------:|:--------:|:------------:|:-----------------:|
| SFT-Baseline       |     -    |   77.7   |    67.2   |  83.5  |   78.7  |     76.8 |     33.9     |        23.2       |
| GPT-4               |    KTO   |   81.0   |    69.0   |  86.0  |   81.1  |     79.3 |     35.2     |        25.8       |
|                    |    DPO   |   81.2   |    70.2   |  86.0  |   81.1  |     79.6 |     36.7     |        25.8       |
| Llama3.1-70B         |    KTO   |   79.4   |    69.9   |  84.8  |    80   |     78.5 |     36.7     |        24.5       |
|                    |    DPO   |   79.4   |    66.2   |  84.1  |   79.3  |     77.3 |     36.1     |        24.5       |
| Llama3.1-405B        |    KTO   |   79.4   |    67.7   |  84.1  |   79.9  |     77.8 |     36.1     |        23.8       |
|                    |    DPO   |   79.2   |    66.9   |  85.4  |   80.5  |     78.0 |     36.6     |        25.5       |
| GPT-3.5-Turbo      |    KTO   |   80.2   |    67.9   |  84.8  |   79.9  |     78.2 |     36.1     |        24.0       |
|                    |    DPO   |   79.7   |    67.7   |  84.8  |   79.9  |     78.0 |     36.1     |        23.8       |
| GPT4o-mini         |    KTO   |   81.2   |    69.2   |  85.4  |   80.5  |     79.1 |     36.7     |        24.0       |
|                    |    DPO   |   80.5   |    67.6   |  85.4  |   81.1  |     78.7 |     36.7     |        25.5       |
| Claude-3-Haiku     |    KTO   |   79.7   |    67.2   |  85.4  |   81.7  |     78.5 |      36      |        24.0       |
|                    |    DPO   |   79.9   |    67.2   |   86   |   81.7  |     78.7 |      36      |        23.5       |

---

### Author Response · Authors · 2024-11-28
**[Revision] Summary of Modifications In Revision**

We thank the reviewers for their constructive comments on improving this work. We have modified the draft to reflect on the issues pointed out by the reviewers and included more experiment results to support the claims. The edited text is highlighted in blue in this version. We summarize the modifications below:
## Major Updates
### Ablation On Test Case Generator
In Appendix A.6, we experimented PLUM across alternative test case generators (e.g., GPT-3.5-Turbo, Llama 3.1) to demonstrate its scalability and cost-efficiency, achieving consistent gains without compromising performance (Table 11). We hope this can help address the comments by **AYWK**  and **Du4Y**.

### Generalization To Stronger Models
In Appendix A.7, we validated the effectiveness of PLUM on stronger models (Qwen-2.5-Instruct/Coder-14B), showing improvements on challenging benchmarks like LeetCode and LiveCodeBench (Table 12). This indicates PLUM works on stronger models (larger and with more advanced post-training). We hope this could help answer Reviewer **AYWK**’s question regarding PLUM’s applicability to more powerful base models.

---

## Clarity Improvements
**Figure 2 Caption:**  Enhanced the clarity of the figure caption to better explain the content and context of the ablation study.

**Table 6 Caption:** Revised the caption for Table 6 to ensure it accurately conveys the table's purpose and findings.

**Algorithm Typo Correction:** Fixed an error in the algorithm body; it now correctly states that “each solution $s_{i,k}$​ passing all test cases are labeled as positive.”

**Improved Layout:** Addressed layout issues, including correcting the order of Tables 3 and 4 to align with the narrative flow.
Additional Related Works: Included relevant references, such as Gee (2023), to provide a more comprehensive discussion of related work.

---

### Note · Authors · 2024-12-16

**Comment:**

We tried our best to answer the queries from all reviewers but be did not receive confirmation regarding whether our responses and contributions were fully acknowledged. Given this uncertainty, we believe it is best to withdraw at this time and continue refining our work.

We sincerely appreciate the valuable feedback and the time the reviewers and committee have dedicated to evaluating our submission.

**Withdrawal Confirmation:**

I have read and agree with the venue's withdrawal policy on behalf of myself and my co-authors.